# Reducing Soil Permeability Using Bacteria-Produced Biopolymer

Amanda Mendonça [1], Paula V. Morais [2,*], Ana Cecília Pires [2], Ana Paula Chung [2] and Paulo J. Venda Oliveira [3,*]

1   Department of Mechanical Engineering, University of Coimbra, 3030-788 Coimbra, Portugal; amaandamendonca@hotmail.com
2   Centre for Mechanical Engineering, Materials and Processes, Department of Life Sciences, Calçada Martim de Freitas, University of Coimbra, 3000-456 Coimbra, Portugal; accp@ua.pt (A.C.P.); apchung@gmail.com (A.P.C.)
3   ISISE, Department of Civil Engineering, University of Coimbra, 3030-788 Coimbra, Portugal
*   Correspondence: pvmorais@ci.uc.pt (P.V.M.); pjvo@dec.uc.pt (P.J.V.O.)

**Abstract:** The building of civil engineering structures on some soils requires their stabilisation. Although Portland cement is the most used substance to stabilise soils, it is associated with a lot of environmental concerns. Therefore, it is very pertinent to study more sustainable alternative methodologies to replace the use of cement. Thus, this work analyses the ability of the more sustainable xanthan-like biopolymer, produced by *Stenotrophomonas maltophilia* Faro439 strain (LabXLG), to reduce the permeability of a sandy soil. Additionally, the effectiveness of this LabXLG is compared with the use of a commercial xanthan gum (XG) and cement for various hydraulic gradients and curing times. The results show that a treatment with either type of XG can be used to replace the cement over the short term (curing time less than 14 days), although a greater level of effectiveness is obtained with the use of the commercial XG, due to its higher level of purity. The soil treatment with LabXLG creates a network of fibres that link the soil particles, while the commercial XG fills the voids with a homogeneous paste.

**Keywords:** biopolymer; soil stabilisation xanthan gum; sandy soil

## 1. Introduction

The growing development and consequent increase in construction in the main urban centres around the world results in a need for infrastructures over and/or inside geotechnical formations lacking the most favourable characteristics. In the past, these formations were frequently considered inappropriate for civil engineering. The main limitations of these types of soils are usually related to insufficient mechanical properties and permeability characteristics that are inadequate for most building practices. Some ground improvement techniques are frequently used to enhance the properties of these soils to overcome these limitations and to allow the safe construction of infrastructures. Currently, chemical stabilisation using cement-based binders is one of the techniques most used [1]. However, the application of cement-based materials raises some concerns from an environmental point of view, in particular: (i) the activity associated with the production of cement creates significant $CO_2$ emissions; in fact, in 2014, this activity contributed to about 8–10% of the global $CO_2$ emissions [2–5]; (ii) the use of cement-based materials for soil stabilisation creates an irreversible composite material [4]; (iii) the use of cement-based binders for soil stabilisation induces an increase in the pH of the soil, with detrimental effects on the vegetation and microbial communities [4].

In order to decrease the ecological footprint caused by the cement-based materials, some alternative bio-based materials that promote the enhancement of the soil's properties have been investigated during the last few years. These biomaterials can be obtained from the activity of microorganisms and/or the use of substances derived from them. Bacteria-producing materials [6–8] and enzymes [8–13] have been used to catalyse the

hydrolysis of the urea in porous media to promote biocementation and biopolymers have been used to modify the properties of soils [5,14]. The effect of the biocementation process on such properties is usually due to the association of two factors, the creation of some bonds in the solid skeleton and the filling of the empty spaces between the soil particles with calcium carbonate crystals. Results published with this methodology using sandy soils have showed that the biocementation process promotes an improvement in the mechanical behaviour [7,9–11,15–17] and a decrease in the coefficient of permeability of the biocemented soil [16,18]. During the last few years, several studies have investigated different biopolymers for soil improvement, specifically in terms of the: (i) improvement of the mechanical behaviour with the use of Guar gum [19], β-glucan [20,21], and xanthan gum [22–26]; (ii) change in the compressibility characteristics with the use of Guar gum [27], cationic e-polylysine [23], xanthan gum [19,23,26], gellan, and agar gums [4,21,28]; (iii) change in the plasticity properties with the use of Guar gum [29], cationic e-polylysine [23], and xanthan gum [22,23]; (iv) reduced permeability coefficient with the use of Guar gum [30], chitosan gum [31], and xanthan gum [20,25]; (v) use of Chitosan gum to remediate contaminated soils [4,32]; (vi) reduce the erosion of the soil with the use of xanthan gum [21] and casein and sodium caseinate salt [31]. Recent research concerning the effect of xanthan gum (XG) on the hydraulic conductivity of XG-soil mixtures can be summarised as follows:

(i)   The biostabilisation of sandy and silty soils with XG induces the filling of the pore spaces of the soil with hydrogels, which obstruct the water flow (i.e., pore-clogging) and consequently reduce the permeability coefficient of the XG-soil mixture. This effect tends to be more significant with an increase in the XG content of the mixture [19,20,25,30,33,34]. In fact, the experimental results of Ayeldeen et al. [34] show that treatment of sand and silty soil with a content of 2% of XG leads to a reduction in the coefficient permeability to 10% of the value shown for the untreated soils.

(ii)  The results of the mixture of XG (0.5%–3%) with sand-bentonite are not in line with the behaviour described previously for biostabilised silty and sandy soils since a slight increase in the permeability coefficient is observed with the increase in the XG content [35]. These results highlight the effect of the finer particles (silt or clay) in the soil type in terms of the effectiveness of the treatment with biopolymers. In fact, the increase in the XG content strengthens and makes more numerous the bonds between the clay particles (ionic/electrostatic, hydrogen, and Van der Waals). This promotes the creation of more and bigger-sized aggregates of clay particles. This leads to the widening of the water flow channels and a consequent slight increase in the coefficient of permeability, within one order of magnitude [35].

(iii) The experimental findings concerning the effect of the curing time on the permeability coefficient of treated soils show some contradictory results. Thus, while Khachatoorian et al. [33] show a decrease in the permeability coefficient of treated Ottawa sand (0.43–0.6 mm size sand) over 11 days of curing, the results obtained by Cabalar et al. [25] with treated Narli sand (0.075–1.0 mm size sand) do not show a clear tendency over the same range of curing time. On the other hand, for a longer curing time, most of the experimental results show an increase in the permeability coefficient with the increase in the curing time for mixtures of XG with sand and silt, which is justified by the dehydration of the XG expected over time [25,34]. Indeed, SEM images confirm the occurrence of shrinkage over time, which increases the size of the pore spaces, and consequently an increase in the soil's permeability [34].

(iv)  The increase in the permeability coefficient observed over the curing time (in the long term) is more significant in treated sands than in treated silts since the finer particles of silt delay the evaporation of the water in the voids and the dehydration of the hydrogels deposited in the pore spaces [34].

(v)   The gas permeability also decreases after the biostabilisation with XG; indeed, a mixture of 3% xanthan gum with kaolin clay decreases the gas permeability of the mixture in relation to the untreated clay by up to two orders of magnitude [36].

This effect is more significant for a higher water content since there is more water available for the hydration of the biopolymers, which promotes a more effective pore-clogging [36].

As the XG is a polysaccharide, there is an important concern, related to its long-term durability when it is used in geotechnical engineering since this biopolymer (like the generality of biopolymers) is biodegradable over time [5]. In spite of this issue, some works have revealed a satisfactory durability, for example: (i) a mixture of sand, gravel, and kaolin treated with XG shows a reasonable durability against water-induced deterioration [37]; (ii) a sand biostabilised with XG submitted to slake durability tests showed a better resistance to disintegration than with the use of Portland cement [38]; (iii) the long-term strength (until 730 days) of a sand treated with XG does not reveal a decrease in the strength, that is, traces of the biodegradability [39]. It should be mentioned that sometimes, the biodegradability of the XG can become an advantage, for instance, in temporary works performed in sensitive areas, from an environmental point of view and where, after the work, it is necessary to remove all traces of the intervention performed.

Although the use of XG in soil stabilization is more sustainable than the use of Portland cement, the industrial production of commercial xanthan gum is not entirely environmentally friendly, since it includes: processes of fermentation, filtration (or centrifugation), precipitation using non-solvents (such as isopropanol, ethanol, acetone), the use of salts, and pH adjustments [40]. In order to mitigate some of the environmental concerns related to the use of commercial XG, this work studies the ability of a xanthan-like gum obtained directly from the strain of *Stenotrophomonas maltophilia* Faro439 (LabXLG) to reduce the permeability of the soil treated. The aim of the present work is to compare the effectiveness of LabXLG with commercial XG and the traditional stabilisation with cement concerning the permeability coefficient of sandy soil. Additionally, the effect of the curing time (3, 7, 14, and 28 days), the hydraulic gradient (6.9, 8.2, 9.4, and 10.6), the XG content (0.16 and 0.33%), and cement (0.33 and 1.0%) are analysed. Scanning electron microscopy (SEM) with energy dispersive X-ray (EDX) was also used to study the microstructure and the chemical composition of the soil treated with LabXLG and commercial XG.

## 2. Materials and Methods

### 2.1. Soil Characterisation

The grain size distribution and the main characteristics of the soil used in the experiments are shown in Figure 1 and Table 1, respectively. This is a non-organic and non-plastic silty sand, composed of 72.7% sand and 27.3% silt, classified as SM (ASTM D2487, 2017) [41]. The soil compaction characteristics reached by using the standard Proctor test (ASTM D698, 2012) Ref. [42] show an optimum water content ($w_{opt}$) of 14.3% and a maximum dry unit weight ($\gamma_{dmax}$) of 16.2 kN/m$^3$. The pH value of the non-treated soil is approximately 7.1 [43].

### 2.2. Production of the Xanthan Gum Biopolymer Compound

The production of the biopolymer enriched compound, illustrated in Figure 2, consisted of the following steps [43]: (i) *Stenotrophomonas maltophilia* strain Faro439 (UC-CCB 127), obtained from the University of Coimbra Bacterial Culture Collection (https://ucccb.uc.pt/, accessed on 15 January 2020), was grown on R2A agar (Reasoner's 2A agar medium) plates for 3 days, at a temperature of 25 °C; (ii) the cells grown were used to inoculate the xanthan medium [44]. The cells were grown in a batch system incubated for 24 h in a shaker at 150 rpm. These cells were used as inoculum in a new batch system with xanthan medium incubated for 3 days in a shaker at 150 rpm before undergoing the centrifugation process to recover the biomass containing the biopolymer compound; (iii) the biomass was drained for 24 h to remove the excess of water and lyophilized for 24 h to obtain the powder form and to facilitate the mixing with the soil.

The use of this xanthan-like, enriched compound is justified by its simple production process. Indeed, the compound is obtained without the use of purification steps, solvents,

or procedures, or any equipment that leads to consequences from the ecological point of view. Therefore, this xanthan gum-like compound production can be considered more environmentally friendly than the commercial XG. Its availability at reasonable prices in relation to other biopolymers [21] and the ability of this biopolymer to decrease the soil's permeability are other positive aspects.

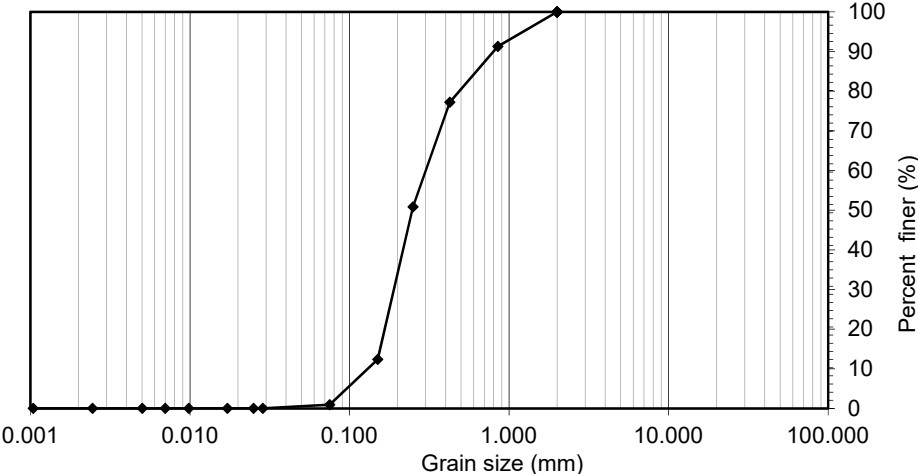

**Figure 1.** Grain size distribution of the soil used.

**Table 1.** Main characteristics of the soil used in the experiments.

| Property | Value |
|---|---|
| Grain size distribution: | |
| Sand (%) | 72.7 |
| Silt (%) | 27.3 |
| Clay (%) | 0 |
| Specific gravity, G | 2.66 |
| Consistency limits | NP (**) |
| Standard Proctor test [ASTM D698]: | |
| Maximum dry unit weight, $\gamma_{dmax}$ | 16.2 |
| Optimum water content, $w_{opt}$ | 14.3 |
| Organic matter content, OM (%) | 0.19 |
| pH | 7.1 |
| Soil classification, USCS (*) [ASTM D2487] | SM |

(*) Unified Soil Classification System; (**) Non plastic.

### 2.3. Commercial Xanthan Gum

The commercial XG used in this work is produced by Sigma-Aldrich(St. Louis, Missouri, EUA) Ref.: G1253), available in powder and is obtained from *Xanthomonas campestris* bacterium via the fermentation of glucose/sucrose. It is a heteropolysaccharide composed of units of glucose, mannose, and glucuronic acid [40]. This is a non-toxic hydrophilic colloid with pseudoplastic rheological properties [40] and is stable within a wide temperature range (10°–80°) and pH (1–13) values. Nowadays, XG is fundamentally used as a thickener and a stabilising agent for many applications, such as agriculture, food, the pharmaceutical industry, cosmetics, industrial lubrication of equipment, textile printing pastes, and explosives, among others [30,40].

### 2.4. Portland Cement

Some mixtures in this research (Table 2) were chemically stabilised with Portland cement Type I 42.5 R, which is composed of 45% of cement particles smaller than 45 μm. The cement is mainly composed of calcium oxide (CaO = 63.0%), which induces a spontaneous reaction with water, i.e., hydraulic properties.

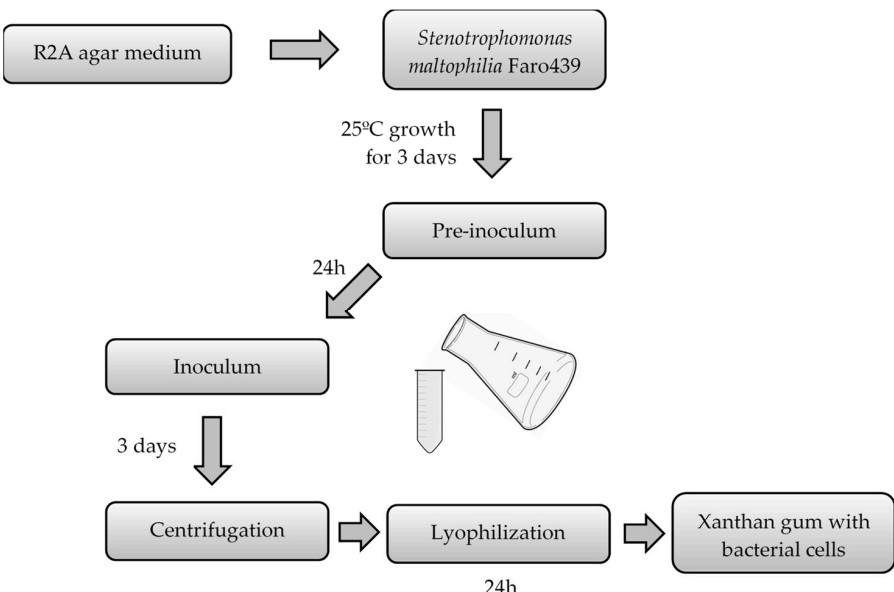

**Figure 2.** Diagram of the production of the *Stenotrophomonas maltophilia* Faro439 strain biopolymer.

### 2.5. Testing Procedure

The samples used in the permeability tests were produced in six different types: natural soil, a mixture of soil with the xanthan-like biopolymer produced by bacteria (Lab XG) (0.33%), a mixture of soil with the commercial biopolymer xanthan gum for the contents of 0.33% and 0.16%, and also a mixture of soil with cement for the contents of 0.33% and 1.0%. Each type of mixture was tested for different hydraulic gradients (6.9, 8.2, 9.4, and 10.6) and after four different curing times (3, 7, 14, and 28 days). The details of the testing programme are shown in Table 2. Although all the specimens were prepared equally, the permeability tests were repeated three times for each condition tested, to assure the quality and the reproducibility of the results obtained.

The stabilised soil specimens used in the permeability tests were prepared according to the following steps: (i) larger particles of the soil were removed with a sieve with a 2.0-mm mesh; (ii) the soil was dried in an oven at 105 °C for 24 h; (iii) the dry soil and powdered grout agent content (presented in Table 2) were mixed; (iv) the soil, the grout agent (LAB XG, commercial XG, and cement), and the water content of 14.3% (optimum water content for untreated soil using the standard Proctor test) were mixed until a homogeneous paste was obtained; (v) the paste was put inside PVC moulds (38 mm in diameter, 76 mm height) in 3 layers; (vi) each layer was compacted in order to obtain the dry unit weight of 16.2 kN/m$^3$ (optimum conditions for untreated soil using the standard Proctor test); (vii) the contact surfaces between two successive layers was scarified before the introduction of a new layer; (viii) the specimens were stored during the curing time (Table 2) in a humid chamber with a temperature of (20 ± 2 °C) and humidity (95 ± 5%) control; (viii) after the curing time, the specimens were assembled in the equipment used to carry out the permeability tests (Figure 3); (ix) a container (to measure the volume of the water that flowed through the specimen) was placed under each one; (x) the water level was adjusted in order to assure the hydraulic gradients stated in Table 2; (xi) the permeability tests were performed.

Although the optimum compaction conditions obtained from the standard Proctor test changed when Portland cement and the xanthan gum were added to the soil [26,45], all the specimens used in the work were prepared for the optimum water content for untreated soil, making it possible to compare all the results without including additional effects.

**Table 2.** Testing programme.

| Grout Agents | | | Hydraulic Gradient (m/m) | Curing Time (Days) |
|---|---|---|---|---|
| **Lab XLG (%) (\*)** | **Commercial XG (%)** | **Cement (%)** | | |
| 0 (\*) | 0 (\*) | 0 (\*\*) | 6.9 | – |
|  |  |  | 8.2 | – |
|  |  |  | 9.4 | – |
|  |  |  | 10.6 | – |
| – | 0.16 | – | 6.9 | 3/7/14/28 |
|  |  |  | 8.2 | 3/7/14/28 |
|  |  |  | 9.4 | 3/7/14/28 |
|  |  |  | 10.6 | 3/7/14/28 |
| – | 0.33 | – | 6.9 | 3/7/14/28 |
|  |  |  | 8.2 | 3/7/14/28 |
|  |  |  | 9.4 | 3/7/14/28 |
|  |  |  | 10.6 | 3/7/14/28 |
| 0.33 | – | – | 6.9 | 3/7/14/28 |
|  |  |  | 8.2 | 3/7/14/28 |
|  |  |  | 9.4 | 3/7/14/28 |
|  |  |  | 10.6 | 3/7/14/28 |
| – | – | 0.33 | 6.9 | 3/7/14/28 |
|  |  |  | 8.2 | 3/7/14/28 |
|  |  |  | 9.4 | 3/7/14/28 |
|  |  |  | 10.6 | 3/7/14/28 |
| – | – | 1.0 | 6.9 | 3/7/14/28 |
|  |  |  | 8.2 | 3/7/14/28 |
|  |  |  | 9.4 | 3/7/14/28 |
|  |  |  | 10.6 | 3/7/14/28 |

(\*) Percentage based on dry weight. (\*\*) Reference tests (non-stabilised soil).

The permeability tests were performed according to the scheme illustrated in Figure 3, which consists of a constant head permeability test. This scheme allows us to test eight specimens simultaneously. An overflow drain in the upper container assures a constant level of water during the tests. As is usual in this type of test, the total volume of water that flows through the specimen is measured (the accumulated water inside the containers placed under each specimen was weighed on a precision balance). The system has a set of taps that make it possible to control the water flow for each specimen. The permeability coefficient (k) was calculated according to Darcy's law:

$$k = \frac{Q}{A \times i} = \frac{(\Delta V / \Delta t)}{A \times (\Delta h / L)} \tag{1}$$

where Q is the water flow rate, $\Delta V$ is the average of the accumulated volume of water that passed through each sample during three measurements, $\Delta t$ is the period of time, A is the cross-sectional area of the specimen, i is the hydraulic gradient, L is the height of the specimen, and $\Delta h$ is the sum of the L/2 and the water column height.

SEM (scanning electron microscope) with EDX (energy dispersive X-ray) were also performed, with the objective of analysing the microscopic structure of the soil-biopolymer mixtures and the chemical elements present in each mixture. The samples were prepared by fixing the soil samples with 2.5% of glutaraldehyde, following by dehydration by a grade of ethanol incubations (70%, 80%, 90%, 95%, and 100%). Previous to observation, a thin layer of gold coating was sputtered on the cracked surface of a thin section of the specimen tested and the SEM/EDS tests were performed in accordance with Pansu and Gautheyrou [46].

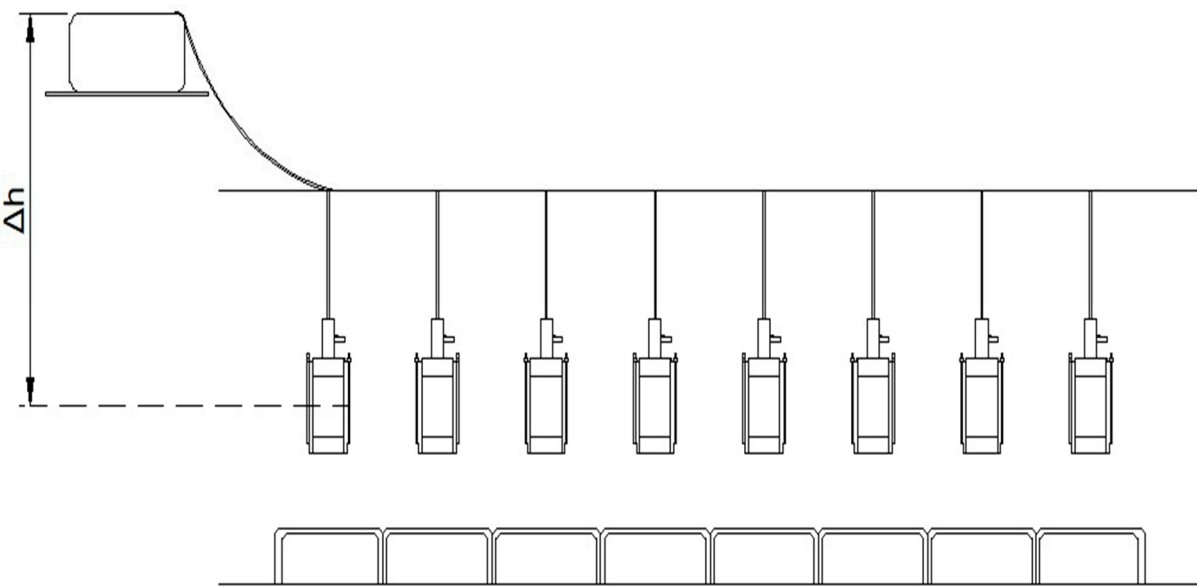

**Figure 3.** Set-up used for the permeability tests.

## 3. Results and Discussion

The permeability coefficients obtained experimentally with the different mixtures and hydraulic gradients are presented in Figure 4a–d for a curing time of 3, 7, 14, and 28 days, respectively. The results from the natural soil are used as reference values to evaluate the impact of each grouting agent on the permeability coefficient of the soil treated. Additionally, the inclusion of the samples chemically stabilised with cement in this work is intended to compare the effectiveness of the two XGs (commercial and the one produced from bacteria) to reduce the soil's permeability, with the most usual grouting agent (cement) in geotechnical engineering.

For a better analysis of the results, the average value, the maximum, and minimum errors for each condition tested are presented. As expected, and as is usual in this type of mixture [7,19], there is a high scattering of the results, which is more significant for the natural soil and the soil treated with Lab XG. In fact, these results reflect some heterogeneity of the natural soil, the lower level of purity of the Lab XG, and the difficulty in producing homogeneous specimens when a low content of XG is used. Indeed, an increase in the content of the commercial XG from 0.16% to 0.33% significantly decreases the scattering of the results.

### 3.1. Effect of Type and Content of the Stabilising Agent

Independently of the curing time and the hydraulic gradient, the results depicted in Figure 4 clearly show that the stabilisation of the soil with both types of XG induces a significant decrease in the coefficient of permeability. This behaviour is due to the development of some links and viscous hydrogels due to the contact of the XG with the water (i.e., hydration of the hydrogels) that fill part of the soil's voids [23,30], inducing a decrease in the soil's permeability. The results also reveal that the commercial XG is more effective than the Lab XG to reduce the soil's permeability; in fact, the use of 0.33% of the LabXLG in the mixtures has much less of an effect than the use of 0.33% of the commercial XG and even with the use of 0.16% of the commercial XG, mainly for higher curing times. This greater effectiveness of the commercial XG is consistent with its greater purity when compared with the LabXLG. Indeed, it is important to state that the LabXLG used in the tests is composed of the lyophilized biomass formed by *S. maltophilia* Faro439, containing the biopolymer and the lyophilized cells that were responsible for producing the biopolymer, consequently a low level of purity is obtained.

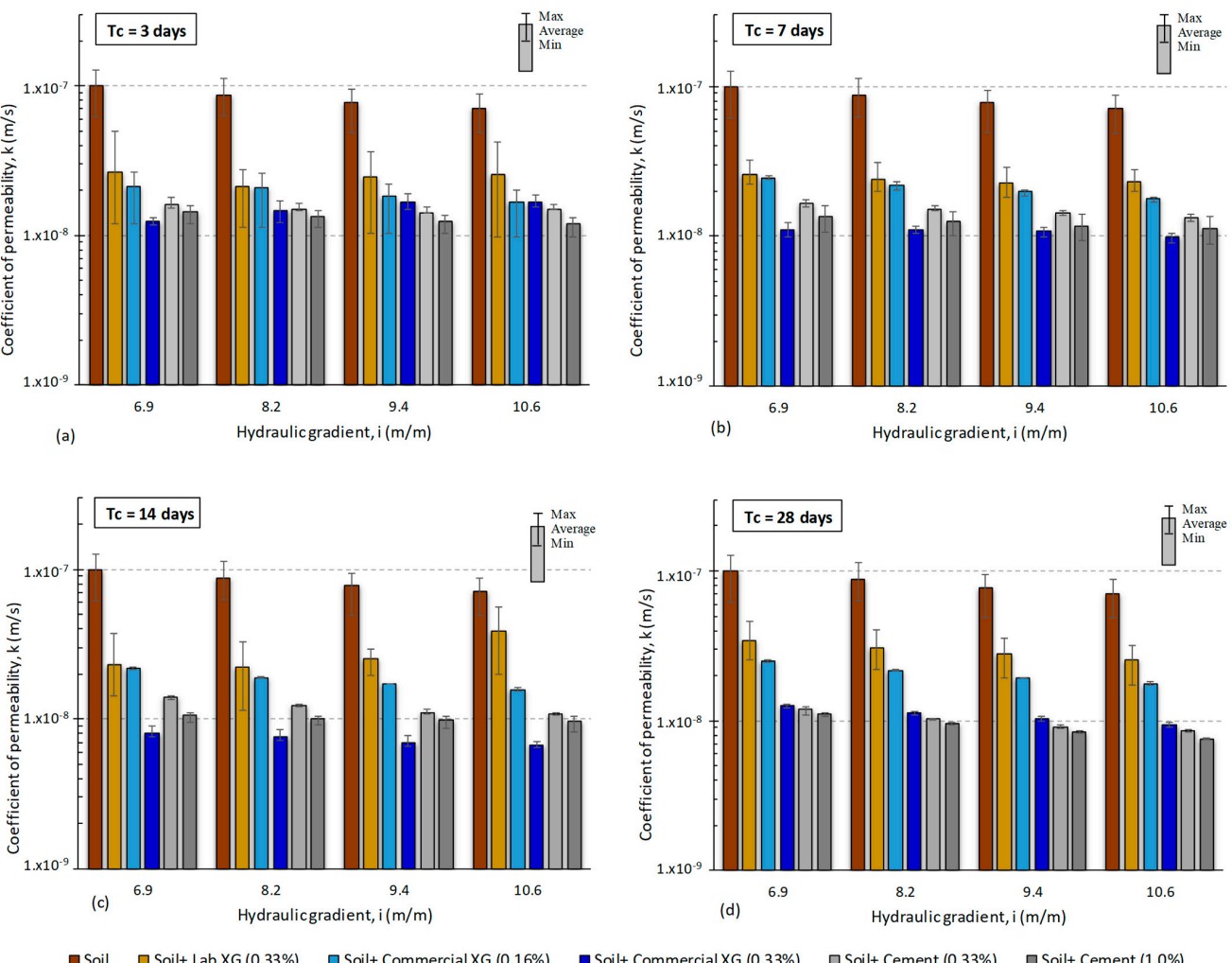

**Figure 4.** Permeability coefficient of the natural soil and stabilised soil for different hydraulic gradients; (**a**) curing time of 3 days; (**b**) curing time of 7 days; (**c**) curing time of 14 days; (**d**) curing time of 28 days.

The comparison of the use of XG with the use of Portland cement (the most usual binder for soil stabilisation) shows that the use of 0.33% commercial XG is more effective than the use of 1% cement, namely for a curing time of 7 and 14 days. Even for a curing time of 28 days, the effectiveness of 0.33% of commercial XG is similar to the use of 0.33% of cement. Thus, these results indicate that XG can be used to replace the use of cement, mainly in the short term (for instance in temporary works).

Considering the efficiency of both biopolymers and that the production process of the commercial XG is associated with a complex industrial process with more environmental concerns (as discussed in Section 1) than the production of the Lab XG, the use of the Lab XG emerges as a viable alternative to reduce a soil's permeability.

### 3.2. Effect of the Hydraulic Gradient

With some exceptions (for instance, the mixture of the soil with the LabXLG for a curing time of 14 days), most of the results show a slight decrease in the permeability coefficient with the increase in the hydraulic gradient, mainly for the natural soil and the treated soil with 0.16% of the commercial XG. Although, this tendency is not in line with most of the results obtained by Cabalar et al. [25] for a range of hydraulic gradient from 5 to 100; in some cases, the increase in the hydraulic gradient from 5 and 10 (similar to the range considered in this work) also shows a decrease in the soil's permeability [25].

The differences previously observed may be due to the methodology used in the tests to evaluate the permeability coefficient. As shown in Figure 3 in the present work, the

permeability coefficient was measured with a descendent water flow. Thus, the increase in the hydraulic gradient tends to increase the vertical effective stress with the consequent decreases in the void ratio, which tends to decrease the soil's permeability, as observed in most of the results in this work. In the case of the specimens being treated with the Lab and the commercial XG, the results also seem to indicate that there may be a movement of some hydrated hydrogels towards the voids in the soil, which tends to restrict the water flow, thereby inducing the decreases in the permeability coefficient.

The results also show that the specimens with a high level of stabilisation (i.e., with a low permeability coefficient) show a lower impact on the variation of the hydraulic gradient. Indeed, the stronger solid skeleton promoted by the stabilisation with a high content of the commercial XG (0.33%) and cement (0.33% and 1%), restricts the eventual movement of the soil particles and hydrogels, which leads to a lower influence of the hydraulic gradient.

### 3.3. Effect of Curing Time

Figure 5 highlights the effect of the curing time on the evolution of the average permeability coefficient over the curing time, for the mixtures with a grout content of 0.33% and subjected to a hydraulic gradient of 6.9. Independently of the process of production of the XG, a sharp decrease in the soil's permeability during the first 3 days of curing is observed, followed by a slight decrease over time to a minimum permeability coefficient for a curing time of 14 days. Although these results indicate that the hydration of the hydrogels tends to occur fundamentally over the short term (<3 days), they also suggest that there is still some hydration of the hydrogels during the first 14 days of curing time. In fact, as the hydration starts from the outside of the hydrogels, this seems to hinder the water penetrating to the inner part of the hydrogels, which leads to a slow and gradual hydration of this inner zone. After 14 days of curing, a slight increase in the soil's permeability is observed with the use of both types of XG, which is probably due to some biodegradation of the XG [14] and/or the dehydration of the hydrogels with the consequent shrinkage of the XG. These two factors increase the pore spaces in the soil over time, which favours the flow of water: consequently, an increase in the soil's permeability is observed. The decrease in the soil's permeability over the short term (fewer than 14 days) is in line with the results of Khachatoorian et al. [33] for a similar range of curing times, while the increase in the permeability coefficient over the long term (more than 14 days) matches the findings of Ayelden et al. [34] and Cabalar et al. [25].

As expected, the specimens chemically stabilised with cement show a reduction in the permeability coefficient with curing time, which is consistent with the development of the pozzolanic reactions that occur in the long term. Thus, for a curing time of 28 days, the coefficient of permeability of the soil treated with cement and the commercial XG with the same content (0.33%) show similar effectiveness, while in the short term, the commercial XG induces a greater reduction in the soil's permeability. Figure 5 also emphasizes the better effectiveness of the commercial XG in relation to the LabXLG, which can be fundamentally attributed to its higher level of purity.

### 3.4. Analysis of the Structures Formed in the Biopolymer-Sol Mixture and Chemical Analysis

The SEM images of the natural soil (Figure 6) show an "open" structure, with a great void ratio that is associated with its high permeability coefficient. Figures 7 and 8 show the SEM images of the structure of the solid skeleton of the soil treated with both types of XG. The use of the LabXLG (Figure 7) seems to induce the creation of structures similar to a network of fibres that link the soil particles [5], which decreases the voids in the soil, thus reducing its permeability. Chang et al. [14] also observed that XG creates bridges between soil particles. However, the use of the commercial XG (Figure 8) seems to produce a homogeneous paste, probably due to the hydration of the hydrogels, which is responsible for the reduction in the permeability coefficient. The differences in the structures of the solid skeleton induced by both types of XG seem to be related to the differences in the

effectiveness of the two XGs to decrease the soil's permeability. In fact, the more open structure promoted by the LabXLG is associated with a higher soil permeability, while the homogeneous paste induced by the use of the commercial XG restricts the water flow, decreasing the soil's permeability in relation to the use of the LabXLG.

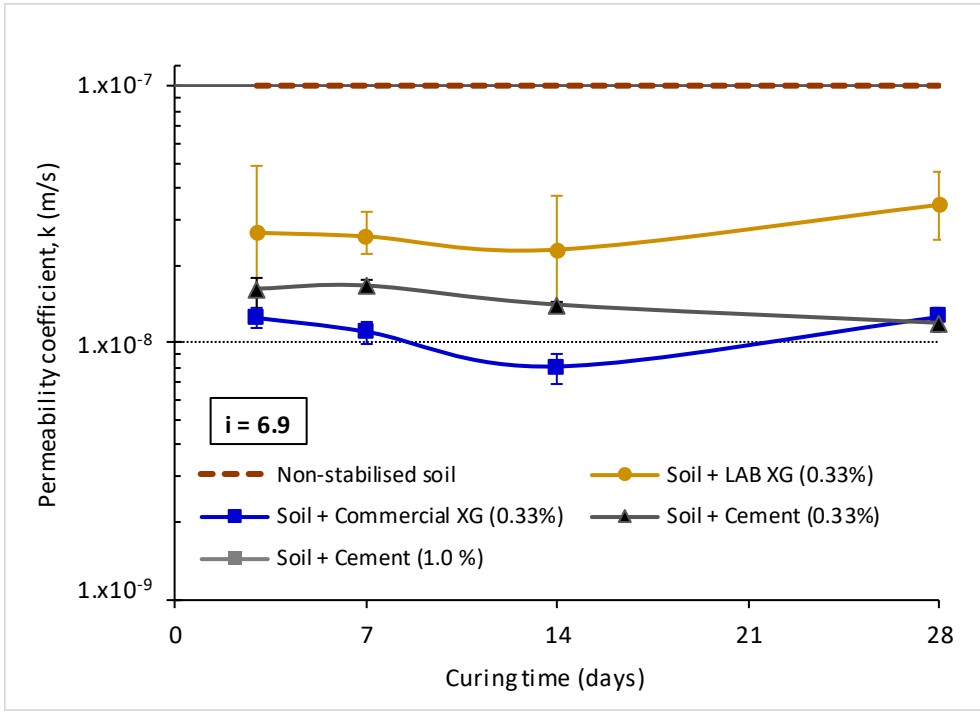

**Figure 5.** Effect of curing time on the permeability coefficient for a hydraulic gradient of 6.9.

Figure 9 shows the chemical analysis carried out on a fibre of the LabXLG, while Figure 9b shows all the chemical elements present in the sample. It is worth mentioning that the gold (Au) must be disregarded because it was used to prepare the specimens for the SEM test. The results obtained show a significant content of carbon (C, 30.8%) and oxygen (O, 22.7%), which clearly indicate the presence of organic fibres [40]. This finding supports the conclusion that biological fibres can be related with the soil permeability reduction.

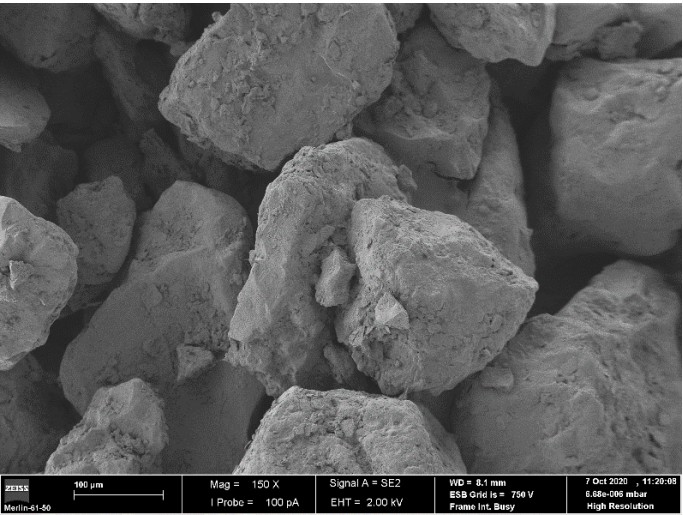

**Figure 6.** SEM image of the natural soil at 150×.

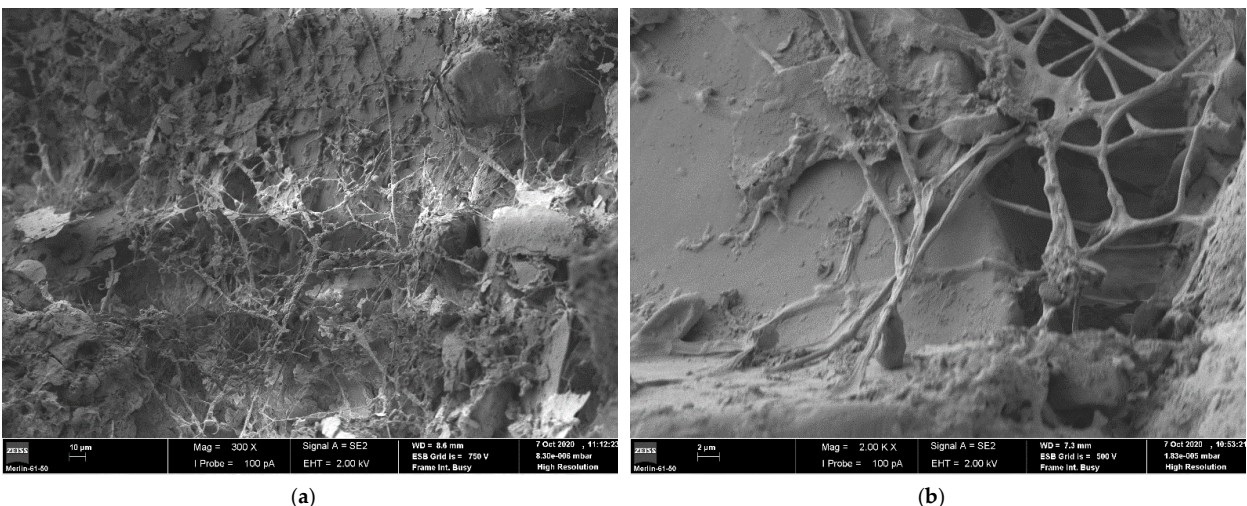

**Figure 7.** (**a**) SEM image of the mixture Lab XG-soil at 300×. (**b**) Scale-up of Figure 7a to 2000×.

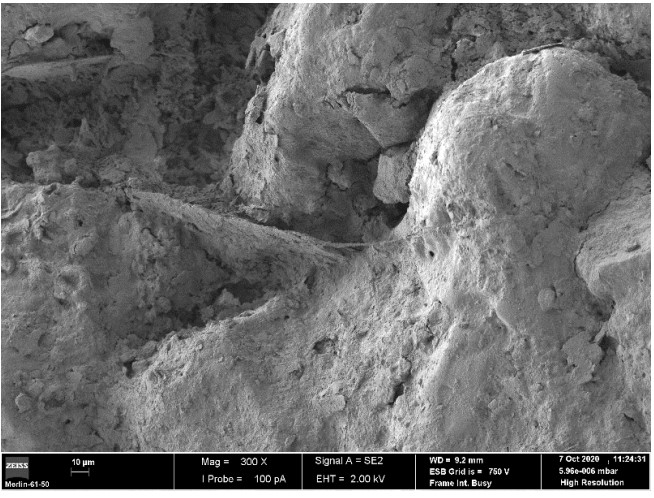

**Figure 8.** SEM image of the mixture produced with the commercial XG at 300×.

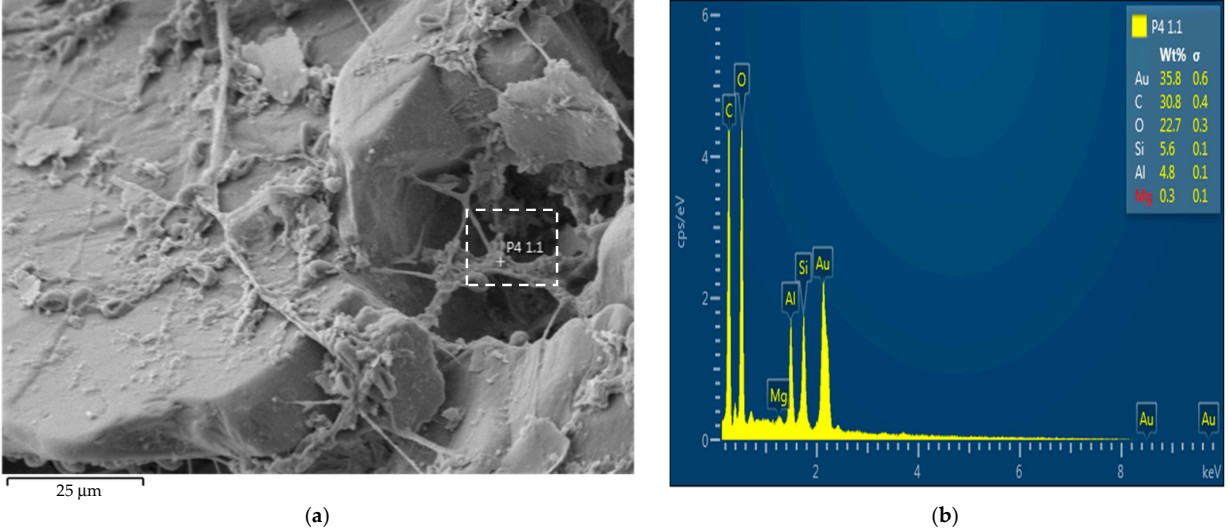

**Figure 9.** (**a**) Area targeted (P4 1.1) for the chemical analysis. (**b**) Chemical analysis of the LabXLG-soil specimen.

## 4. Conclusions

This work compares the ability of the xanthan-like gum obtained directly from a strain of *S. maltophilia* Faro439 species (LabXLG), with commercial xanthan gum (XG) and Portland cement to decrease the permeability coefficient of a sandy soil. In parallel, the effect of the curing time (3, 7, 14, and 28 days), the hydraulic gradient (6.9, 8.2, 9.4, and 10.6), and the XG content (0.16 and 0.33%) and cement (0.33 and 1.0%) are also analysed. From the experimental testing programme, the following conclusions may be drawn:

(i)   The treatment of the soil with both types of XG induces a significant decrease in the coefficient of permeability compared to the untreated soil.

(ii)   The comparison of the two types of XG shows that the LabXLG is less effective than the commercial XG, which is related to its lower level of purity. The lyophilized biomass formed by *S. maltophilia* Faro439 also contains lyophilized cells as well as the biopolymer.

(iii)   The use of a 0.33% content of commercial XG is more effective to reduce the coefficient of permeability of the treated soil than the use of 1% cement, namely for a curing time of 7 and 14 days. These results indicate that XG can replace the use of cement in the short term, for instance in temporary works.

(iv)   The increase in the hydraulic gradient induces a slight decrease in the permeability coefficient. This is consistent with the descendent water flow used in the permeability tests since the increase in the hydraulic gradient tends to increase the vertical effective stress with the consequent decreases in the void ratio and the soil's permeability.

(v)   The treatment with both types of XG demonstrates a slight decrease in the permeability coefficient during the first 14 days of curing, followed by an increase for longer times. The initial decrease in the soil's permeability is associated with the hydration of the hydrogels, while the increase in the soil's permeability over time probably reflects the existence of some biodegradation of the XG and/or the eventual dehydration of the hydrogels associated with the consequent shrinkage.

(vi)   The microstructure of the treated soil depends on the type of XG, which is linked with each one's effectiveness to reduce the soil's permeability. Thus, the LabXLG creates a network of fibres that links the soil particles and decreases the voids in the soil, while the commercial XG induces the partial filling of the pore spaces with a homogeneous paste, probably due to the hydration of the hydrogels.

Finally, it should be mentioned that taking the production process and its effectiveness in reducing the permeability coefficient into consideration, the experimental results of this work show that the xanthan gum produced using the *S. maltophilia* Faro439 strain emerges as an interesting alternative for use in the stabilisation of sandy soils.

Although *Stenotrophomonas maltophilia* is included in the BSL-2 facultative pathogen group, there are no regulations available for the release of this type of microorganism into the environment, and the number of cells detected in the assays with soil is very low. Therefore, future studies are still needed to ensure that this material is fully safe.

**Author Contributions:** Conceptualization, A.C.P., P.V.M. and A.P.C.; writing—original draft preparation, A.M., A.C.P. and P.V.M.; writing—review and editing, A.M., P.V.M., P.J.V.O. and A.P.C.; funding acquisition, P.V.M. and P.J.V.O. All authors have read and agreed to the published version of the manuscript.

**Funding:** This work was supported by the projects ERAMIN2 REVIVING funded by Fundação para a Ciência e Tecnologia (FCT), project PTDC/CTA-AMB/31820/2017 and FCT UIDB/00285/2020, BIORECOVER H2020 grant agreement 821096, project POCI-01-0145-FEDER-028382 and the R&D Unit Institute for Sustainability and Innovation in Structural Engineering (ISISE), under reference UIDB/04029/2020.

**Institutional Review Board Statement:** Not applicable.

**Informed Consent Statement:** Not applicable.

**Data Availability Statement:** No new data were created or analysed in this study. Data sharing is not applicable to this article.

**Acknowledgments:** This work had the support of the lab technician J.A. Lopes, Department of Civil Engineering, University of Coimbra.

**Conflicts of Interest:** The authors declare no conflict of interest.

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
