# Peer review of "Reducing Soil Permeability Using Bacteria-Produced Biopolymer"

_applsci, doi:10.3390/app11167278_

Round 1
Reviewer 1 Report
The manuscript of Mendonca and coworkers contains an interesting application of bacterial „biopolymers”, namely the use of them as a soil permeability reducing agent. The field is worth investigating, and there are lots of examples in the literature. Unfortunately, the manuscript in question – however contains some interesting data – is scientifically incomplete, important questions remain untouched. The manuscript describes the skeleton of a real experiment, but this is not enough. There are lots of open questions which need to be answered to achieve a state of ready for publication manuscript. This manuscript is not ready for publication. Moreover, the questions regard deeply the experimental part, thus the manuscript in its present form must be rejected.
Let us go one by one, and see the critical points.
- The title is strange for a microbiologist. Aside from the typist’s error („sustanable”) the term bacteria has to be capitalized.
- Reading the abstract, the 5th row presents the first important problem. Apart from the nomenclaturally incorrect name usage (the correct name of the bacterium is Stenotrophomonas maltophilia), this microbe is a BSL-2 facultative pathogen. Thus its applicability for environmental release is questionable. Moreover, we get no information about the origin of the strain, there is no characterization of it, even the produced biopolymer is not characterized, etc. Thus we do not know the truth of the statement in the penultimate sentence of the abstract. What is behind? Really the structure differs, or only the chemical differences influence the swelling parameters?
- The introduction part reviews „biostabilization” attempts. Unfortunately, it lacks the in depth analysis of the inconsistencies of the results. Weather the inconsistencies result from the use of different xanthan polymers, from the use of different methodology or what. It is a mere description, thus does not help to understand why this novel approach would be important. There is one more problem with it. The xanthan biopolymer can be degraded. This limits its applicability as a construction aid. Why is this approach not introduced at all, though mentioned latter in the discussion part as a possibility?
- In the materials and methods part soil characterization refers to data in Figure 9. However, in Fig 9 the elemental analysis targets a „biopolymer” filled space, and not the untreated soil.
- Concerning the biomass production the „xanthan medium” is neither described, nor at least referred with literature.
- The environmental release of a BSL-2 microbe biomass is questionable!
- The argumentation on the „environmentally friendly” production of the biopolymer in question is simply a hype without any scientifically sound argumentation. The authors did not investigate the real footprint questions.
- The exact origin and type of the „commercial XG” must be described! Without that it is unclear what is used as comparison.
- The description of the testing procedure seems to be very precise. However, the key issue is the water content of the soils. The authors argument, that they used 14,3 % because it is optimum for the „Proctor test”. But our concern must be rather the optimum water content for the applied agent, and the goal of the „stabilization agent application”. This question is not at all discussed! Lots of other experiments had to be done to test the ideal water content the get the best stabilization effect!
- The SEM preparation parameters are not described.
- The third paragraph of the Results and discussion part clearly describes why the water content optimization experiments ought to be done...
- Unfortunately the argumentation in chapter 3.1. is not at all based on the chemical structure/nature of the polymers. That is why the type and origin of the commercial one has to be added, and the characterization of the laboratory product has to be performed. Whiteout that everything here is mere speculation.
- Figure 4 has to be redesigned. The legends over each bar type graph are unnecessary, it is needed just once; we do not know what the purpose of the „cloud bar” in the upper right corner is.
- The argumentation again in favor of „Lab XG” as a green alternative is a hype.
- The argumentation around curing time effect uses the possibility of XG biodegradation. This must be examined e.g. experimentally, but at least discussed with references!
- The SEM images presented in the discussion parts have all different magnification / resolution. This is deceptive. Moreover, the text refers to such figures (Figure 11, and 10a) which do not exist. Why we do not see the bacterial cells in case of Lab XG treated samples?
- The conclusions simply repeat the results. But the argumentation is false in case the knowledge on the structure, chemical nature of the polymers is missing.
Author Response
In attachment I send the reply to Reviewer #1

Reviewer 2 Report
This article presents a comparison between the effect of LabXG, commercial XG, and cement on the permeability of a sandy soil. The reviewer is not convinced that the results of this study are significant since the main conclusion is that commercial XG performs better than the proposed LabXG. The reviewer recommends detailed cost and/or environmental footprint analyses to potentially show the benefit of using LabXG over commercial XG.
Other detailed comments:
- Several typos exist in the manuscript including one in the title of the article. Please, proofread the article carefully before resubmission.
- One major question about the work is the cost to scale up the production of the proposed LabXG versus the very cheap cost of commercially available XG. Providing that the permeability of the soil with the two XGs is the same, the cost should be analyzed carefully.
- Page 2 of 15: Missing references on the use of Xanthan Gum with clay fillers. These are directly related to the topic of this article and need to be cited as part of the literature.
- Page 3 of 15: Table 2 is mentioned in the text before Table 1. Consider removing Table 2 from the first place it was called. Also, Figure 9 in the "soil characterization" section should change to Figure 1.
- Testing Procedure Section:
- Is the percentage used based on dry weight or volume?
- The optimum moisture content referred to in this section is for the untreated soil. There is enough proof in the literature that cement and XG change the optimum moisture content of a soil. While the authors used the same initial moisture content in their samples, they need to comment on the potential effect of missing the optimum point for the treated samples on the results.
- Page 7 of 15:
- Avoid paragraphs made of a single sentence.
- Can the authors estimate the percentages of the different components in the LabXG? And therefore use the percentage of the LabXG that corresponds to the percentage of the pure commercial XG? This will help to confirm that the difference is due to the high purity level of the commercial XG compared to the LabXG.
- How did the authors prepare the samples for the SEM images? These images are surficial images and thus the technique used to prepare them must be described somewhere in the article.
- Figure 10 and Figure 11 were mentioned in the Article but they are not provided. This makes reading the last part of the article very confusing.
- Why the chemical analysis of the Lab XG-soil is important? The authors just listed the findings of this analysis and presented the chemical analysis results in Figure 9.b without mentioning why should the reader care about this analysis.
Author Response
In attachment I send the reply to Reviewer #2

Round 2
Reviewer 1 Report
The revised manuscript of Mendonca and coworkers - however improved - is still insufficient for publication. First it is simply repeating many former studies. The novelty concerns only the applied methodology, i.e. the xanthan-like polymer was produced by a bacterial strain isolated earlier by the authors. Unfortunately the authors did not answer the biosafety question, how to use, or release a level 2 facultative pathogen as a construction agent? Moreover it is not only a facultative pathogen, but concerning the performance criteria it seems not to be better then the commercially available materials. The authors argue at the use of their "invention" by the smaller footprint and price of production. Unfortunately this has not been investigated and presented in the manuscript. Thus, it is an empty statement.
The authors similarly hypothesize biodegradation and shrinkage of their material / agent. Unfortunately this has not been investigated either. Why the authors did not investigate these facts, if they conclude that their "new agent" could be used for temporary interventions.
There are some minor mistakes also in the text.
- The Portland cement used was "composed ... of cement particles smaller then 45 mm". Is this dimension correct?
- A double dot in the last sentence of the manuscript is not necessary.
Reviewer 2 Report
Thank you for adequately addressing my comments.
Author Response
The authors thank the reviewer for their comments which enhance the quality of the manuscript.
Round 3
Reviewer 1 Report
Unfortunately the authors totally misunderstand the biosafety concern. They used lyophilization not simple "drying". This will not kill the cells, rather transforms the liquid into a more infectious form. Imagine to work with hundreds of litres or hundreds of grams of a BSL 2 organism in the construction industry. Nobody will give a license for such a product.
Apart from the biosafety concerns, still there is practically no novelty in the study. Xanthane gum has already been used for soil stabilization purpouses. The only novelty seems to be that in spite of a purified "chemical" a crude biomass with unknown properties (e.g. cell mass/ EPS ratio, what is the exact composition of the EPS) will be applied.
Author Response
| Please, find in attachment |
